# Brain Metabolic Alterations in Alzheimer’s Disease

**DOI:** 10.3390/ijms23073785

**Published:** 2022-03-29

**Authors:** Carlos G. Ardanaz, María J. Ramírez, Maite Solas

**Affiliations:** 1Department of Pharmacology and Toxicology, University of Navarra, 31008 Pamplona, Spain; cgarcia.41@alumni.unav.es (C.G.A.); mariaja@unav.es (M.J.R.); 2IdISNA, Navarra Institute for Health Research, 31008 Pamplona, Spain

**Keywords:** glucose, lactate, astrocyte, GLUTs, hypometabolism, astrocyte–neuron lactate shuttle (ANLS), neurodegeneration

## Abstract

The brain is one of the most energy-consuming organs in the body. Satisfying such energy demand requires compartmentalized, cell-specific metabolic processes, known to be complementary and intimately coupled. Thus, the brain relies on thoroughly orchestrated energy-obtaining agents, processes and molecular features, such as the neurovascular unit, the astrocyte–neuron metabolic coupling, and the cellular distribution of energy substrate transporters. Importantly, early features of the aging process are determined by the progressive perturbation of certain processes responsible for adequate brain energy supply, resulting in brain hypometabolism. These age-related brain energy alterations are further worsened during the prodromal stages of neurodegenerative diseases, namely Alzheimer’s disease (AD), preceding the onset of clinical symptoms, and are anatomically and functionally associated with the loss of cognitive abilities. Here, we focus on concrete neuroenergetic features such as the brain’s fueling by glucose and lactate, the transporters and vascular system guaranteeing its supply, and the metabolic interactions between astrocytes and neurons, and on its neurodegenerative-related disruption. We sought to review the principles underlying the metabolic dimension of healthy and AD brains, and suggest that the integration of these concepts in the preventive, diagnostic and treatment strategies for AD is key to improving the precision of these interventions.

## 1. Introduction

The evolutive success of the human brain relies on the development of higher cognitive functions tightly linked to its neocortical expansion, density and complexity. These upgraded brain capacities enable humans to achieve an augmented degree of complexity regarding behavior, choice and processing of emotional and social context. Nonetheless, the processes underlying these functions are coupled to a significantly elevated energy demand. In this context, the field of brain energetics is still enigmatic, lacking answers to some fundamental questions aimed at understanding how brains are fueled: How are energy production and energy demand coupled? How do neurons and glial cells participate in this coupling? Is energy handling at the core of neurodegeneration?

The central nervous system (CNS) shows unique metabolic features with a relatively high resting metabolic rate and local, activity-dependent fluctuations, differing from other tissues. This notion is based in two main observations: (a) representing only 2% of total body mass, the brain accounts for roughly 20% of the oxygen and 25% of the glucose consumed by the human body, which implies an exceedingly high metabolic rate [1,2]; and (b) brain glucose utilization is increased upon local brain activity [2]. The high energetic demand exhibited by the brain is required for processes as crucial as the restoration of the ion gradients that allow for and are challenged by electrical signal transmission, or for neurotransmitter uptake and recycling [3,4,5,6,7]. Hence, the maintenance of brain function presumes an elevated metabolic cost, especially considering the billions of incessant synaptic transmissions that account for up to 80% of the total energy requirement ascribed to the functioning of the neuronal network [5]. In this line, when ATP production does not fulfill energy requirements (situations such as aging, ischemia and/or neurodegenerative disorders), synapses appear to be more vulnerable to dysfunction and degeneration [6].

Focusing on specific brain cell types, neurons are responsible for 70–80% of total brain energy consumption, while the remaining portion (20–30%) of energy is expended by glial cells, i.e., astrocytes, oligodendrocytes and microglia [6,8], illustrating that the metabolic signature differs between different brain cells. Understanding cell-specific energy metabolism and intercellular metabolic interactions within the brain is fundamental to elucidate the groundings of brain function and adaptation to environmental changes. Moreover, several studies have evidenced that metabolic alterations could be at the center of the onset and progression of many neurodegenerative diseases. Aging processes induce decreases in the glucose and oxygen metabolic rates of brain cells [9], and this metabolic breakdown seems to be further exacerbated in degenerative disorders such as Alzheimer’s disease (AD), Parkinson’s disease (PD), Huntington’s disease (HD) and amyotrophic lateral sclerosis (ALS) [10].

## 2. Brain Energy Fuels

Glucose is the main energy source for the adult brain [1,2], although in certain circumstances the usage of other energy sources is also possible. One of the alternative energy sources that has drawn attention in recent years is lactate. In the past, lactate was ignored, as it was considered only a metabolic dead-end substrate. However, this view started to change completely when several authors demonstrated how lactate represents an energy fuel for the brain: (i) infused lactate is preferred over glucose by human brains [11], (ii) lactate is metabolized via the tricarboxylic acid cycle (TCA) within the human brain [12], (iii) plasma lactate can be metabolized by neurons to an extent similar to glucose [13], and (iv) lactate aids neurons in meeting their energy requirements [14].

Apart from lactate, ketone bodies (3-β-hydroxybutyrate (3HB) and acetoacetate (AcAc)) can also be used by brain cells as an energy substrate. Specifically, during brain development, the energy supplied by ketone body metabolism fulfills 30–70% of the energy requirements, thereby making ketones essential during this period [15,16,17]. This idea is reinforced by rodent studies demonstrating that enzymes involved in ketone body metabolism are elevated during lactation, and drastically diminish after weaning [18,19]. In fact, the use of ketone bodies is crucial during these first stages of life in order to support the high metabolic demand, as well as lipid and amino acid synthesis, required for brain development [20,21]. In the adult brain, ketone body metabolism is increased in conditions in which brain glucose availability is compromised, such as during fasting, starvation, low carbohydrate/high-fat intake and prolonged exercise, while reliance on ketone bodies drops greatly during fed states [22,23]. Under fasting conditions, the liver is the organ responsible for ketone body generation from fatty acids and amino acids [24].

Despite being one of the most energy-demanding organs in the body, the brain mostly lacks intrinsic energy reserves. Therefore, it appears to be highly dependent upon the continuous supply of blood-borne energy substrates. Perturbing or interrupting this process leads to serious neurological function impairment, loss of consciousness and coma after a few minutes. The supply of essential nutrients, ions and signaling molecules, while preventing the indiscriminate access of solutes into the brain, is guaranteed by the existence of the blood–brain barrier (BBB) and the blood–cerebrospinal fluid barrier (BCSFB), which tightly control the flux of molecules between the blood and cerebral fluids. The BBB is composed of capillary endothelial cells interconnected by tight and adherent junctions, their underlying basement membrane, pericytes, and astrocytic endfeet. As noted above, the BBB orchestrates the entrance of metabolites such as glucose, amino acids and ketones from the circulation into the CNS. Importantly, the BBB impedes the access of blood-borne immune cells, toxins and pathogens that could interfere with adequate neuronal function [25].

Depending on their chemical attributes, different solutes pursue different strategies to cross the BBB. Small lipid-soluble molecules, as well as oxygen and carbon dioxide, undergo passive transport, diffusing down the concentration gradient. However, polar molecules and large peptides, such as proteins, hormones, growth factors and neuroactive peptides [26,27,28], require interaction with receptors or transporters to overcome the tight junction restriction erected by the BBB [29,30,31,32,33]. The transporters present in the BBB can be classified into two main families, depending on their need to consume energy, i.e., hydrolyze ATP, to transport the solute: (i) the ABC family, corresponding to ATP-binding cassette proteins, which hydrolyze ATP to transport solutes and (ii) the solute carrier (SLC) family, corresponding to solute carrier proteins that exert facilitated transport without ATP expenditure. The function of the ABC transporters is to move lipid-soluble molecules across the BBB against their concentration gradients, requiring ATP expenditure to achieve this. The function of the SLC family, composed of more than 300 members, is to maintain a continuous influx of carbohydrates, amino acids, monocarboxylic acids, nucleotides, fatty acids, and organic anions and cations to the brain [33].

Brain energy metabolism is heavily dependent on certain SLC transporters, namely glucose transporters (GLUTs) and monocarboxylate transporters (MCTs) (Table 1). In short, the supply of energy to the brain takes place in the following manner: the entry of arterial blood-borne glucose into the brain is mediated by GLUT1. This transporter is highly expressed by BBB endothelial cells and astrocytes, which present a 55 kDa and a 45 kDa GLUT1 isoform, respectively. Once in the brain, glucose can enter neurons via GLUT3, which is mainly localized in axons and dendrites. Due to the high glucose affinity presented by GLUT3, neurons can receive a continuous glucose supply even when interstitial glucose concentration becomes scarce. Although GLUT1 and GLUT3 are the main glucose transporters in the brain, other members of the GLUT family can be found in specific cell types or brain areas. For instance, GLUT4, which is an insulin-dependent glucose transporter, can be found in neurons, astrocytes and endothelial cells [34]. Indeed, neuronal GLUT4 is fundamental to maintain the energy rates of active synapses, concretely sustaining action potential firing [35]. It has been demonstrated that certain hypothalamic glutamatergic neurons express GLUT2, which acts as a brain glucose sensor and favors sugar-seeking behavior upon hypoglycemia [36]. Microglial cells show relatively abundant expression of GLUT5, which is a fructose carrier with very low glucose affinity [37]. Neurons can also express GLUT6 [38], while GLUT7 has been detected in astrocytes [39]. Finally, GLUT8 has been detected in the neurons of certain brain regions such as the hippocampus, the amygdala, the cerebellum and the hypothalamus [40,41].

Focusing on the predominant glucose carriers in the brain, i.e., GLUT1 and GLUT3, it is remarkable that both proteins are constitutively located in the cell membrane, implying that they do not translocate upon insulin stimulation. This important feature suggests that brain glucose uptake is not insulin-dependent. The role of both GLUT1 and GLUT3 has been extensively studied employing knockout (KO) mouse models: heterozygous GLUT1 KO mice exhibit reduced brain volume and alterations in motor behavior [42]. Of note, this mouse model reproduces the phenotypic changes observed in human patients with GLUT1 deficiency [43]. Moreover, heterozygous GLUT3 deletion in mice induces alterations in social behavior and perturbance of spatial learning and working memory [44].

Besides SLCs, BBB endothelial cells also express sodium-dependent unidirectional transporters that belong to the Na^+^-coupled glucose transporter (SGLT) family (Table 1). SGLT1 and 2 couple the sodium electrochemical gradient to transfer glucose against its concentration gradient across the membrane. According to the literature, although their role in physiological conditions is still unknown, they appear to acquire a central role in brain metabolism upon oxygen/glucose deprivation or ischemia [45].

Regarding MCTs, 14 different MCTs with different substrate affinities have been described (Table 2). Among them, BBB cells express MCTs 1–4, which are responsible for lactate, ketone bodies and pyruvate proton-linked transport. Specifically, MCT1 is present in BBB endothelial cells [15], astrocytes [46], oligodendrocytes [47], and microglia [48], and shows high affinity for pyruvate, although it can also transport lactate and ketone bodies. In neurons, the main monocarboxylate transporter is MCT2 [49], which shows a higher affinity than MCT1 for all of their substrates [46]. MCT3 is present in retina and choroid plexus and can only transport lactate [50]. Finally, MCT4 is the astrocyte-specific lactate carrier [51]. These distribution and differential affinity patterns suggest that MCTs have a central role in energy fuel shuttling within different brain cell types.

## 3. Astrocyte–Neuron Metabolic Coupling

Within the human brain, the number of astrocytes is significantly higher than the number of neurons [52]. Astrocytes are essential for many functions in the CNS, such as glutamate homeostasis, glycogen storage, water homeostasis, tissue repair, release of gliotransmitters (that modulate neurotransmission), synapse formation and synapse remodeling, among other processes [53,54].

The unique astrocytic cytoarchitecture grants these cells a privileged position to sense microenvironmental fluctuations and accordingly elaborate dynamical responses. Astrocytes possess two different processes: fine processes that ensheath synapses, and larger vascular processes in contact with blood vessels (named “endfeet”) [55,56,57]. A wide range of receptors for neurotransmitters, cytokines and growth factors, as well as various transporters and ion channels, can be found in perisynaptic processes. Astrocytic glutamate receptors are particularly interesting, as they operate as sensors of glutamatergic synaptic transmission. Astrocytic endfeet also express several proteins, including important transporters and channels such as GLUTs and aquaporin 4 [55]. 

Considering the features mentioned above, it is tempting to speculate that astrocytes are ideally positioned to sense and respond to neuronal activity, granting an appropriate brain energy supply through their endfeet, which enwrap blood vessels. Indeed, as neurons lack direct contact with blood vessels, many essential substrates such as blood-borne glucose and oxygen interact with astrocytes before accessing neurons [58]. In this line, it is now widely accepted that astrocytes play a central role in the spatiotemporal coupling between neuronal activity and increased cerebral blood flow, i.e., the so-called neurovascular coupling [59].

Astrocytic and neuronal populations express different patterns of metabolism-regulating genes, resulting in divergent energy substrate preference [60,61,62]. Accordingly, neurons and astrocytes present different (but complementary) metabolic profiles, paving the way for extensive metabolic cooperation.

Neurons need to maintain high energetic rates; therefore, they rely mainly on energy-efficient oxidative metabolism to meet their energy needs [13,63,64,65]. Notably, neurons can use lactate as their energy substrate [66,67,68,69,70].

Glucose enters neurons via GLUT3, and once inside the cell is phosphorylated by hexokinase (HK), resulting in glucose-6-phosphate (G6P) subsequently directed to the pentose phosphate pathway (PPP) and the glycolytic pathway (Figure 1). The final product of glycolysis is pyruvate, which after entering the mitochondria will be subjected to the TCA cycle, and subsequently to oxidative phosphorylation consisting of the electron transport chain (ETC). This process consumes O_2_ and leads to the production of ATP and CO_2_. G6P undergoing PPP is transformed into 6-phosphogluconate (6PG), and is thereafter converted to ribulose-5-phosphate (R5P). In this process, nicotinamide adenine dinucleotide phosphate (NADPH) is produced, an essential molecule to regenerate oxidized antioxidants, e.g., glutathione (GSH) and thioredoxin. Energy storage in the form of glycogen is not possible within neurons, as these cells lack glycogen synthase (GS), which is constitutively degraded via glycogen synthase kinase 3 (GSK3) phosphorylation and subsequent ubiquitin-dependent proteasomal digestion.

Compared to neurons, astrocytes are highly glycolytic and exhibit significantly lower rates of oxidative metabolism [61,64,71]. In astrocytes, glucose is imported via GLUT1, and most of the glucose subjected to glycolysis is converted to lactate and subsequently released to the extracellular space [60,64,65,69,72] (Figure 1). The high glycolytic preference and the higher tendency for the production of lactate over pyruvate is due to the selection of specific genes involving enzymes and transporters that favor this phenotype: for example, lactate dehydrogenase (LDH) is the enzyme that catalyzes the interconversion of glycolysis-derived pyruvate into lactate, or vice versa. Interestingly, astrocytes express the LDH5 subunit, which favors the formation of lactate from pyruvate. Therefore, instead of conveying glycolysis-derived pyruvate to oxidative metabolism, astrocytes prefer to convert it into lactate. Contrarily, neurons lack LDH5 and express the LDH1 subunit, which favors the formation of pyruvate from lactate [73]. Upon receival of exogenous lactate, neurons will tend to convert it into pyruvate, which can subsequently undergo oxidative metabolism as explained above. These results support the notion of an astrocytic preference for a glycolytic, lactate-producing phenotype and of a neuronal preference for an oxidative, lactate-consuming phenotype.

Moreover, astrocytes are capable of storing glucose in the form of glycogen (Figure 1). Despite its relatively low level in the CNS compared to peripheral tissues, glycogen is the largest energy reserve within the brain. Glycogen is located throughout various astrocytic subcellular compartments such as their cytosol, endfeet and perisynaptic processes [74,75]. Glycogen represents an advantageous form of glucose storage because its metabolization is genuinely fast. In addition, glycogenolysis has the advantage of producing G6P without ATP consumption and can yield ATP under anaerobic conditions [74,75]. 

The evidence of a markedly different astrocytic and neuronal metabolic profile suggests that brain energy metabolism is a compartmentalized process. Notably, the metabolic phenotypes of these two cell types appear to be largely complementary and intimately coupled. With respect to neurons, astrocytes take up significantly higher quantities of glucose. Indeed, it can be considered that astrocytes obtain disproportionally high amounts of glucose compared to their energy requirements. How can this attribute be linked to the fact that the most energy-requiring brain cell type is the neuron? A possible explanation for this paradox could be the existence of an energy substrate transfer from astrocytes to neurons. This idea was introduced more than two decades ago by Pellerin and Magistretti [72], who described the astrocyte–neuron lactate shuttle (ANLS). According to this model, upon neuronal activation, extracellular glutamate levels increase and the released glutamate is taken up by astrocytes, via specific glial glutamate transporters. As this process is Na^+^-dependent, the resulting increase in intracellular Na^+^ concentration activates the Na^+^/K^+^-ATPase, increasing ATP consumption, glucose uptake, and glycolysis in astrocytes [76]. The elevation of the glycolytic rate leads to the production of a high amount of lactate, which is subsequently released to the extracellular space and can be taken up by neurons as energy fuel for oxidative phosphorylation-derived ATP production [77,78] (Figure 1).

The idea of an energetic coupling between astrocytes and neurons is also supported by the fact that glycogen is found in astrocytes and not in neurons. The first demonstration that glycogen metabolism may involve an astrocyte–neuron interaction was obtained by Magistretti and co-workers, who showed that the release of neurotransmitters from the neuron could induce glycogenolysis in astrocytes [79,80]. The importance of glycogen stores in the preservation and viability of neuronal activity upon glucose scarcity, such as hypoglycemia, also reinforces the idea of the interaction between astrocytes and neurons involving glycogen [75,81]. Of note, astrocytic metabolic requirements may also be fulfilled by glycogen mobilization [82,83]. Indeed, glycogen breakdown induces an elevation in lactate production and release [83,84], contributing to the fulfillment of neuronal energy needs (according to the ANLS model).

## 4. Brain Energy Metabolism in Aging

Aging is an irreversible and intricate process characterized by the time-dependent functional decline of physiological integrity [85], thereby being the main risk factor for susceptibility to neurodegenerative diseases [86]. In 2020, globally, there were 727 million individuals (9.3% of the world’s population) aged 65 years or over. Over the next three decades, the number of older individuals worldwide is projected to more than double, reaching over 1.5 billion (16% of the world’s population) by 2050 [87].

The senescent brain shows a plethora of alterations intimately related to the aging process. However, most of these changes are not completely understood at the molecular level. Redox environment alterations, low-grade chronic inflammation and deficits in brain bioenergetics are defined as major contributors to the cognitive decline associated with aging. In this line, 20–40% of healthy people between the ages of 60 and 78 years declare having problems with working, spatial and episodic memory, as well as with processing speed [88,89]. In contrast, other types of major cognitive functions do not decline until very advanced stages of senescence, such as semantic memory, or might even present life-long stability, such as emotional processing [90]. Importantly, these cognitive alterations show a close correlation with age-induced neuroanatomical changes [91].

The brain suffers a marked decrease in energy metabolism during senescence [9]. In this line, the glucose hypometabolism and mitochondrial dysfunction measured by neuroimaging techniques are described as age-related early indicators [92,93,94]. Interestingly, a decline in glucose utilization has been observed in different brain areas through fluorodeoxyglucose positron emission tomography (FDG-PET) analyses [95]. Focusing on concrete brain regions, an age-related metabolic decline has been observed in both parietal and temporal cortices, and it is especially accentuated in the frontal cortex [96]. Studies performed in rats have revealed that the learning and memory impairment can be directly linked to the age-dependent reduction of glucose utilization in the hippocampus and prefrontal cortex [97]. Unfortunately, functional brain imaging studies lack the sufficient resolution to determine a clear temporal sequence regarding brain energy decline and structural changes. However, several authors have suggested that the elevated mitochondrial capacity and oxidative metabolism that facilitated brain expansion during evolution could also underlie the cognitive deficiencies correlated with brain senescence [98,99,100,101,102,103].

Senescence-related brain hypometabolism may be linked to several factors. An inverse correlation between aging and cerebral blood flow has been observed in clinical studies [104,105]. Moreover, BBB permeability increases in aged people compared to young subjects [106]. These two factors together, i.e., brain hypoperfusion together with lower BBB integrity, lead to impaired nutrient import as well as toxin removal. Furthermore, blood-derived proteins, such as fibrinogen, immunoglobulins, albumin, thrombin or hemoglobin, as well as peripheral immune cells could access the CNS, inducing neuroinflammation [107]. 

Based on human as well as animal studies, the aging brain shows a marked decrease in GLUTs [108,109], as well as changes in the expression of enzymes involved in oxidative phosphorylation and glycolysis [110,111,112]. For example, a study performed in Fischer rats showed that brain glucose uptake is decreased with age, and that this is correlated with decreased expression of neuronal GLUTs. Specifically, GLUT3 and GLUT4 expression falls dramatically in the aging brain, while the decrease in the expression of endothelial GLUT1 (55 kDa) is not so pronounced [113]. Paradoxically, the expression of astrocytic GLUT1 (45 kDa) appears to be increased with age, which suggests a metabolic modification in astrocytes and neurons [114]. In contrast, in another study, performed in Wistar rats, aging induced a decrease in astrocytic GLUT1 expression compared to newborn astrocytes, suggesting a reduction in glucose transport from the blood into the astrocytes [115]. This same result has been observed in 15-month-old mice, where the decrease in GLUT1 was accompanied by reductions in GLUT3 levels [108]. A strong argument supporting the causal role of brain hypometabolism in the development of cognitive deficits is the demonstration that GLUT1 KO mice develop an age-dependent decrease in cerebral capillary density, reductions in blood flow and subsequent decrease in glucose uptake, together with BBB leaking [116,117]. Remarkably, all of these metabolic changes occurred before hippocampal spine loss and behavioral impairments [116]. 

Apart from the alterations regarding glucose transporters, aging also induces a decline in MCT1 expression [108]. This phenomenon, together with the reduced expression of other transporters relevant to the astrocyte–neuron metabolic coupling, might be a key characteristic of age-related brain metabolic disturbances. In this line, imaging studies have shown decreased neuronal mitochondrial metabolism and increased glial mitochondrial activity in the elderly [70]. Neuronal mitochondrial disturbances lead to decreased ATP levels, lower glycolytic rates and increased oxidative stress, together with neuronal apoptosis [118,119]. The presence of damaged mitochondria is a common feature in aging that correlates with the accumulation of damaged and dysfunctional organelles [85,120]. This issue can be especially harmful for neurons with long unmyelinated axons, numerous synaptic connections and high energy rates, such as the neurons present in the cortex, hippocampus and basal ganglia [121,122,123,124]. Regarding astrocytes, upregulated mitochondrial metabolism may be an energy-saving mechanism to satisfy energy requirements under stress conditions. Based on previous studies, the metabolic shift from anaerobic metabolism towards mitochondrial respiration that occurs in astrocytes correlates directly with age [125]. Of note, this metabolic shift would produce lower rates of lactate, disrupting the ANLS and subsequently depriving neurons of one of their main energy sources. Overall, enhanced mitochondrial metabolism and decreased lactate levels may at least partially explain the state of metabolic insufficiency in the senescent brain.

## 5. Energy Metabolism Alterations in Alzheimer’s Disease

Neurodegenerative diseases are characterized by neuronal dystrophic structural changes and loss of function. At the subcellular level, different neurodegenerative conditions such as AD, PD, ALS or HD share certain features including abnormal protein aggregation, protein degradation impairment, axonal transport failure, mitochondrial dysfunction and cell death [126]. Although the etiology of these diseases is still largely unknown, many authors agree that reduced energy metabolism, excitotoxicity and oxidative damage are at the center of their pathogenesis [127,128,129,130]. PET imaging studies have demonstrated a drastic reduction of the glucose signal within the affected regions of patients suffering from AD, PD, ALS or HD [10]. Specifically, patients with mild cognitive impairment (MCD) precedent to AD show a significant reduction in glucose uptake in the entorhinal cortex and parietal lobes. These deficits become more pronounced and anatomically more widespread during AD pathology progression. Remarkably, the anatomical distribution of deficient brain glucose metabolism serves as a clinical tool to distinguish AD from frontotemporal dementia (FTD), PD, Lewy body disease and other types of dementia [131,132,133].

AD is the most common form of dementia, accounting for up to 60–70% of all cases. According to a 2021 report from the Alzheimer’s Association, deaths due to this disease between 2000 and 2019 have increased 145%. Considering the improvement in life expectancy and the aging of the population, these numbers are expected to increase. For this reason, the high prevalence of AD is one of the major global health issues [134,135]. 

AD can appear at an early age (<65), but the large majority (90–95%) of patients have a late onset. Although the etiology of the disease is still unknown, the pathological hallmarks are well-described. AD is characterized by the presence of amyloid plaques, made up of extra-cellular amyloid-β (Aβ) peptide, and neurofibrillary tangles, composed of intracellular hyperphosphorylated tau protein. Aβ is generated from the amyloid precursor protein (APP) by sequential cleavages mediated by BACE-1 and the β-secretase complex [136,137]. The degree of Aβ clearance has a great role in its accumulation [138]. Indeed, healthy Aβ levels in the brain depend on the balance between synthesis, re-uptake and clearance. 

In a very simple manner, AD can thus be considered as a proteinopathy where pep-tides are deposited in brain structures that are critical for memory and cognition. Consequently, it induces loss of neuronal functions, alteration of synaptic connections and neuronal cell death in different brain regions [139]. Among all of the affected areas, the hippocampus is considerably one of the most important ones, as it is widely accepted that declarative memory, which includes spatial representations, is selectively dependent on a network centered on the hippocampus [140].

Currently, there is no effective treatment or cure for AD. Several drugs can ameliorate symptoms and retard cognitive decline for up to 6–12 months, but none of them stops its progression. Moreover, most new AD therapies have failed to revert the symptoms, even if they can remove Aβ plaque accumulation in the brain [136,141,142,143].

In recent years, brain energy deficiency and metabolic alterations have attracted attention suggesting that AD could be a bioenergetic disease. Indeed, studies performed in human subjects suggest that high blood pressure, obesity, diabetes and atherosclerosis are risk factors for AD [144]. All of these pathologies imply energy metabolism alterations and/or brain vasculature modifications leading to reduced neuronal energy supply, which contributes to increased vulnerability to cognitive decline. Thus, in light of the observed alterations in brain glucose uptake and metabolism as well as mitochondrial bioenergetics, AD can be classified as a metabolic disease [145]. Activity-dependent regional glucose uptake and metabolism have been extensively analyzed by FDG-PET studies [146,147,148,149,150,151,152]. Reductions in brain glucose metabolism measured by FDG-PET scanning take place years before the emergence of clinical AD symptoms [153] and also precede decreases in hippocampal volume within the brains of cognitively healthy patients with increased genetic risk for AD [154]. At early stages of the disease, impairments in glucose metabolism are mainly detected in the parieto-temporal and posterior cingulate cortex, and are extended to other frontal areas during the progression of the disease. Primary visual and motor cortices appear to be less severely affected, while basal ganglia, thalamus and cerebellum are relatively spared [155,156,157]. Very interestingly, the locus coeruleus (LC) is one of the areas that degenerates with disease progression in AD [158,159,160,161,162]. The LC releases noradrenaline (NA), which subsequently induces astrocytic intracellular Ca^2+^ and cAMP elevations, resulting in several cellular responses including aerobic glycolysis enhancement [161,163]. Hence, degeneration of the LC could contribute to the alteration of glucose metabolism in AD [164]. 

In this context, diminished brain glucose utilization is accompanied by a shift to-wards a more ketogenic metabolism [108]. When the brain reaches a ketogenic state, the main fuel shifts from glucose to favor ketone bodies, such as 3HB and AcAc. It has been extensively addressed that ketogenic diets induce beneficial effects in epilepsy, as they can inhibit glutamatergic excitatory transmission [165], and it is postulated that these diets could also mitigate neuronal hyperexcitability that occurs in AD [166]. This hypothesis is supported by the evidence of decreased brain glucose metabolism [167] together with a maintained ketone body metabolism that does not decline in any of the stages of AD pathology [168]. Furthermore, mitochondrial biogenesis is promoted by ketone bodies, and synaptic functions seem to be stabilized by a ketogenic diet [169]. Ketone bodies have also shown the ability to reduce the generation of reactive oxygen species and to increase ATP availability [170]. 

Moreover, it has been extensively demonstrated that BBB integrity is compromised in patients with AD [107,171,172,173,174]. Indeed, age-induced BBB permeability is exacerbated in mild cognitive impairment or early stages of AD compared to neurologically healthy individuals [175,176], suggesting that neurovascular dysfunction is an early event in the pathogenesis of AD. Additionally, AD brains show changes in nutrient transporters and metabolic enzyme expression level and activity. For instance, AD patients exhibit lower levels of brain GLUT1 and GLUT3 [177,178], especially in the dentate gyrus of the hippocampus [178] and the cerebral cortex [179], which correlate with decreased brain glucose uptake and cognitive deficits [180]. In AD patients, GLUT1 and GLUT3 reductions have been related with tau hyperphosphorylation and reduced levels of HIF1α (that leads to the transcriptional activation of both GLUTs) [181]. Eighteen-month-old APP/PS1 mice, an AD mouse model, exhibited reduced hippocampal GLUT1 levels compared to wild type group, whereas no differences were found in younger animals (8 months old) [182]. In another AD model, 3xTg mice, a decline in GLUT3 and GLUT4 was also described [183], coinciding with alterations in brain glucose uptake [108]. The brain glucose hypometabolism observed in AD brains promotes synapse loss, neuronal death, energetic deficits and neurotoxic protein accumulation, which mutually aggravate one another in a vicious cycle [131,184,185,186]. 

Apart from changes in glucose transporters, other members of the SLC family also appear to be altered in AD. Specifically, the members of the SLC1 and SLC17 glutamate transporter families altogether encompass two subclasses, i.e., the excitatory amino acid transporter (EAAT) subclass and vesicular glutamate transporter (VGLUT) subclass, that are responsible for the regulation of glutamate homeostasis and, therefore, could affect excitotoxic neuronal injury in neurodegenerative diseases such as AD [187]. Astrocytes express EAAT1 (*SLC1A3*) and EAAT2 (*SLC1A2*), which transport 5% and 90% of the glutamate in the CNS, respectively [188,189]. Studies performed in postmortem AD brains have revealed pathology-specific EAAT2 splice variants and concomitant glutamatergic dysfunction at early stages of the disease [190]. In this line, the presence of an excess in the amount of extracellular glutamate is considered a risk factor for AD [191]. Indeed, a metabotropic glutamate receptor antagonist, i.e., memantine, is effective for the symptomatic treatment of AD. Therefore, EAATs might be an intriguing therapeutic target [191]. However, studies performed in AD mouse models have obtained conflicting results concerning EAAT2 expression. For example, in transgenic mice that overexpress mutant human APP, EAAT1 and EAAT2 protein expression and function were found to be diminished in the brain [192]. In a similar manner, another study found decreased expression of EAAT2 protein in the cortex and hippocampus, leading to a drastic alteration of glutamate reuptake activity [193]. In contrast, in another two AD mouse models, i.e., APP/PS1 and 3xTg-AD, EAAT2 protein expression was intact over the pathology of the mice [194]. The lack of good methods for qualitative analysis can possibly underlie these conflicting results, and further research is needed to verify these findings and develop new compounds that could activate EAAT protein expression, providing a beneficial effect in AD patients.

In recent years, the presence of SGLTs in the mammalian CNS has been extensively proved [195]. SGLT1 is found in areas such as the hippocampal CA1, CA3 and dentate gyrus subfields, while SGLT2 has been identified not only in the hippocampus but also in other areas such as cerebellum and also in BBB endothelial cells [196,197,198,199]. Although the role of these transporters in AD pathology is not deeply studied, the particular distribution of SGLTs may be responsible for the intriguing evidence suggesting their neuroprotective properties [200]. Indeed, recent reports propose that SGLT inhibition might alleviate the AD pathogenic process. The first study that addressed the effect of the treatment with an SGLT2 inhibitor, empagliflozin, reported a reduction of soluble and insoluble Aβ levels and senile plaque density in the cortex and hippocampus of APP/PS1 mice treated with empagliflozin [201]. Moreover, the study showed beneficial effects regarding cognitive function assessed by the novel object recognition test and the Morris water maze, suggesting that empagliflozin treatment positively affected cognitive abilities and significantly ameliorated memory impairment. 

Inadequate neuronal glucose as well as mitochondrial energy supply compromise misfolded protein clearance from the brain. Conversely, abnormal protein accumulation promotes mitochondrial damage, decreases energy production and improves oxidative stress [186,202]. Of relevance, mitochondrial dysfunction is particularly important in AD [203,204]. Lack of damaged mitochondrial clearing by mitophagy further exacerbates bioenergetic breakdown of vulnerable neuronal circuits in AD [205]. Mitochondrial metabolism alterations have been analyzed in rats using [^1–13^C]-glucose via ^13^C and ^1^H NMR spectroscopy, and a decline in the incorporation of glucose-derived ^13^C into glutamate, glutamine, aspartate, and GABA has been discovered in the aging brain [206]. In humans, ^13^C and ^1^H NMR studies in healthy subjects have demonstrated that neuronal mitochondrial metabolism is ~30% lower in elderly subjects compared to young [70,207]. In contrast, the astrocytic TCA rate was ~30% higher in the elderly compared to young subjects, indicating that normal aging induces a decrease in neuronal metabolism together with an increase in glial metabolism that can be further accelerated in AD brains [125]. Indeed, given the role of astrocytes as brain energy suppliers, the observation that progressive perturbations affecting astrocytic metabolism and functionality are associated with the disruption of brain glucose metabolism in AD patients is especially relevant [154]. Thus, one could argue in favor of a central contribution of astrocytes to AD-related abnormalities in brain glucose metabolism. In fact, exposition to Aβ oligomers triggers glucose hypometabolism in human stem cell-derived astrocytes [208]. In this line, a series of studies suggest a glutamate-related disruption of the ANLS in AD. According to the ANLS, astrocytic glutamate transport triggers astrocytic glycolysis, and AD astrocytes show decreased glutamate uptake [209]. Interestingly, astrocytes surrounding Aβ plaques also show impaired glutamate transport in vivo [210], and Aβ-induced disruption of this transport disrupts synaptic transmission [211]. Astrocytic glutamate transport has also been demonstrated to drive the FDG-PET signal [212], which is markedly diminished in AD. Finally, deletion of the astrocytic glutamate transporter GLT-1 is sufficient to accelerate the progression of AD in mice [213].

As mentioned before, one of the key characteristics of AD is the accumulation of Aβ, which exacerbates brain glucose hypometabolism both directly within the area of Aβ accumulation as well as in remote regions, probably due to capillary blood flow constriction [186]. In turn, this hypometabolic state triggers inflammatory processes and cellular damage [186,214]. Moreover, alterations in glial cell function together with tau phosphorylation and subsequent accumulation exacerbate aging-induced metabolic changes [215] and Aβ and/or phosphorylated tau-induced network hyperexcitability (Figure 2). Together, these features perpetuate a vicious cycle of neurodegeneration and declining brain glucose metabolism [216,217] that contributes not only to the deterioration of memory and cognition but also to abnormal behavior in affected patients.

Another key feature in AD is neuroinflammation. Of note, neuroinflammation can be facilitated by the high levels of microglial activation that are linked to neurodegenerative conditions [218]. Dramatic changes have been observed in aged microglia, which exhibit altered morphology, together with decreased motility accompanied by slower mobility toward the injured site [219,220]. Moreover, aged microglia show altered responses to extracellular ATP signals compared to young microglia, being less dynamic and branched during senescence [219]. Degenerated neurons appeared to be surrounded by activated microglia in aging and neurodegenerative diseases [221]. Notably, senescent microglia also undergo metabolic reprogramming, shifting their glucose preference to fatty acids for energy production. This profound metabolic change can be directly linked to oxidative stress and inflammation [222]. In AD, Aβ can induce microglial activation through NF-κB, which leads to the elevation in the production of pro-inflammatory cytokines. Moreover, cerebral glucose hypometabolism has been linked to microglial activation in the early onset of AD [223]. Treatment of microglial cells with plasma from AD patients induced a reduction in mitochondrial respiration and increased glycolysis [224]. Indeed, AD-plasma-treated microglia showed diminished NAD^+^ levels, increased LDH release, and decreased oxygen consumption rates with subsequent extracellular acidification. Therefore, blood circulating components from AD patients induce a shift in microglial energy metabolism. The subsequent NAD^+^ depletion could be a fundamental feature in the pathology of AD [225]. In this line, supplementation with NAD^+^ alleviates neuroinflammation, decreasing the number of activated microglia and increasing brain anti-inflammatory cytokine (IL-10) release [226]. NAD^+^ has also been proposed to regulate neuroinflammation by inhibiting the major inflammasome NLRP3, a key component of the innate immune system, highly involved in AD pathology [227]. 

One of the metabolic agents that is receiving increasing attention among geneticists performing AD studies is apolipoprotein E (ApoE), which has been demonstrated to play a major part in conditioning an individual’s risk of developing AD. In humans, ApoE can adopt three different isoforms: ApoE2, ApoE3 and ApoE4. Subjects that are homozygous and heterozygous carriers of ApoE4 respectively show 12 and 2–3 times increased risk of developing AD [228]. Crucially, brain hypometabolism has been unequivocally linked to ApoE4 [229,230,231]. ApoE is mainly produced by astrocytes, and its main role is the transport of cholesterol involved in lipid metabolism, ensuring lipid delivery to neurons and thus neuronal structure maintenance and injury repair [232,233,234]. ApoE can also be expressed by neurons under stressful conditions [235]. For all of these reasons, it is not surprising that different ApoE isoforms heavily influence brain metabolism. In fact, the possible link between ApoE4 and human brain glucose uptake has been extensively studied [236,237,238]. Indeed, mouse models that carry the ApoE4 human allele show decreased metabolic gene expression as well as diminished cerebral glucose uptake compared to ApoE3 carriers [239,240,241]. Moreover, at the cellular level, ApoE4 expression in astrocytes induces alterations in glycolysis, glucose uptake and lactate secretion [238,241]. 

Apart from the above-mentioned brain energy alterations, central insulin resistance is now considered an intrinsic characteristic of AD [242,243,244,245,246]. Although insulin resistance in the periphery is assumed as a classical sign of type 2 diabetes mellitus [247,248,249], the focus is now on the possible existence of central insulin resistance in AD [250,251,252]. Indeed, lower levels of insulin, C-peptide, and insulin growth factor (IGF) levels are detected in the cerebrospinal fluid (CSF) of AD patients [253,254,255,256]. Abnormalities in insulin and/or IGF receptor levels (mRNA or proteins) or activity have also been observed in postmortem AD brains, accompanied by alterations of the downstream cellular signaling cascades [252,254,255,256,257,258,259]. Hence, strategies aiming at preserving or increasing brain glucose metabolism or insulin signaling are of central interest regarding AD.

## 6. Brain Energy Metabolism as Treatment Opportunity for Alzheimer’s Disease

Currently, several clinical investigations are focusing on the improvement of brain energy metabolism as a therapy for neurodegenerative disorders (Figure 3). Brain metabolism modulation can be achieved either in a direct manner with treatments such as intranasal insulin, or indirectly, acting on peripheral metabolism with diet modifications (low-fat, ketogenic) or insulin modulators [260]. Within this last group of insulin modulator drugs, some of them have shown promising results in the treatment of AD, especially metformin [261,262], thiazolidinediones [263,264,265,266], SGLT2 inhibitors [267] and glucose like peptide-1 (GLP-1) analogues [268,269,270,271,272]. 

A detailed summary of the cognitive, metabolic, cellular and proteinopathy-alleviating effects of these proposed strategies can be found in Table 3. Although the beneficial effects obtained by these drugs have shed a bit of light and hope for the development of future AD therapies, current results should be interpreted with caution, as there have been no new drug approvals for the treatment of dementia so far [273,274]. Thus, the lack of effective therapeutic strategies for dementia encourages the deep study of non-pharmacological or pharmacological approaches that improve metabolic functions in an attempt to find the definitive cure for AD.

Any future directions shaping the development of a treatment for AD centrally depend on optimizing brain energetics, as the brain requires sufficient energy resources to undergo the energetically expensive processes needed for cognitive maintenance and/or recovery, such as synaptic regeneration, peptide clearance or proper fueling of electrical signaling [260]. Without fulfilling these process-limiting energy requirements, the benefit of any treatments enhancing such processes could be severely hindered, as the brain would lack the basic resources to undertake them. Accordingly, validating therapeutic or co-therapeutic agents with an energy-ensuring objective, such as GLP-1 analogues or SGLT2 inhibitors, which improve brain glucose metabolism and/or insulin sensitivity, could be promising to unravel the full potential of current AD therapies. 

Another future direction to explore is the combination of pharmacological therapy together with diet and lifestyle interventions, given that this combination has demonstrated a potentiated relief of AD clinical symptoms [313]. In this line, low-carbohydrate ketogenic diets have demonstrated the ability to improve memory and cognition [299,300,301,302,303,304,305,306,307,308,309,310,311,312], and the physical exercise-derived hormone irisin is known to abrogate both synaptic and memory deficits in AD mouse models [314]. Last, but not least, urgent questions regarding a metabolic-integrating AD diagnosis arise: can different FDG-PET abnormal profiles predict different disease progressions? Can other neuroimaging techniques, such as ketone PET, be combined with FDG-PET to more precisely describe the individual metabolic abnormalities of a singular AD patient? And importantly, can these biomarkers predict whether a certain intervention is delaying or reverting disease progression? Provided that neuroimaging-detectable alterations in brain glucose metabolism appear in AD brains before the onset of clinical symptoms, characterizing the pattern of brain FDG-PET alterations (i.e., areas affected, evolution) emerges as an important way to personalize specific AD diagnoses and design targeted interventions accordingly.

## 7. Concluding Remarks

This review summarized the complexity of brain energy metabolism, highlighting the complex cellular and molecular mechanisms involved. The initial neuron-centric view of brain bioenergetics is evolving into a more integrated vision where astrocytes and neurons collaborate in a complementary way. The importance of this astrocyte–neuron metabolic coupling is illustrated by a plethora of evidences showing that disturbances in this orchestrated process can contribute to neurodegeneration.

Impaired brain bioenergetics appears to be a central hallmark of neurodegenerative diseases, preceding the onset of clinical symptoms in AD. Metabolic alterations occur at multiple levels, including reduced neuronal glucose uptake, impaired glycolysis and TCA cycle, all of which adversely impact axonal transport, mitochondrial function and ATP production. In this sense, a provocative question can be raised: in the light of the multiple energetic pathways that seem to be involved in AD pathology, should pharmacological approaches target a single and specific receptor, protein or enzyme? Taking into account the variety of factors altered in brain energy metabolism in AD, this approach seems to be clinically ineffective, and multimodal intervention could be more effective for a successful brain energetic rescue. Overall, maintaining and guaranteeing the energy status of the brain should become a cornerstone for trials attempting to delay the onset and progression of AD.

## Figures and Tables

**Figure 1 ijms-23-03785-f001:**
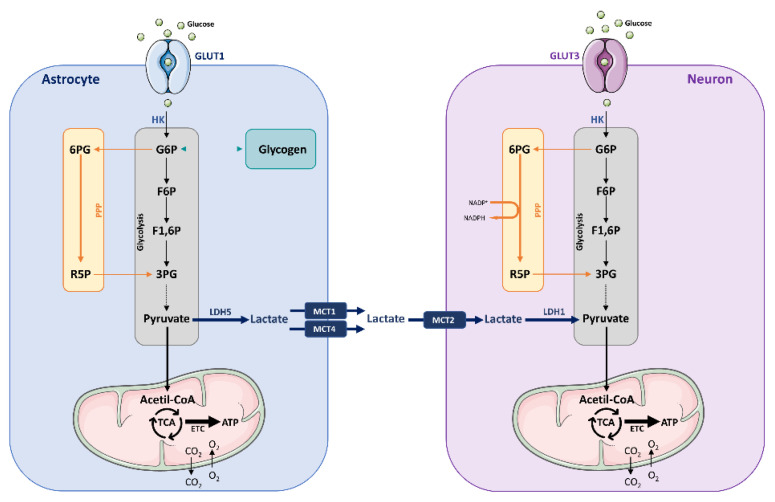
Different metabolic strategies used by neurons and astrocytes. In astrocytes, glucose is imported through glucose transporter 1 (GLUT1) and preferentially stored as glycogen, or metabolized via glycolysis. The generated pyruvate is converted to lactate thanks to the expression of lactate dehydrogenase 5 (LDH5). Glucose enters neurons via GLUT3, and once inside the cell is phosphorylated by hexokinase (HK), resulting in glucose-6-phosphate (G6P) subsequently directed to the pentose phosphate pathway (PPP) and the glycolytic pathway. The final product of glycolysis is pyruvate, which after entering the mitochondria will be subjected to the tricarboxylic acid (TCA) cycle, and subsequently to oxidative phosphorylation consisting of the electron transport chain (ETC). This process consumes O_2_ and leads to the production of ATP and CO_2_. G6P undergoing PPP is transformed into 6-phosphogluconate (6PG), and is thereafter converted to ribulose-5-phosphate (R5P). In this process, nicotinamide adenine dinucleotide phosphate (NADPH) is produced, an essential molecule to regenerate oxidized antioxidants. The weak glycolytic activity of neurons may reduce pyruvate formation and, therefore, obtain limited energy production in mitochondria from glucose metabolism. However, this may be compensated by the uptake of lactate from astrocytes, given that glutamate stimulates lactate release from astrocytes (astrocyte–neuron lactate shuttle; ANLS). Abbreviations are as follows: 3PG, 3-phosphoglycerate; 6PG, 6-phosphogluconate; ETC, electron transport chain; F6P, fructose-6-phosphate; F1,6P, fructose-1,6-diphosphate; G6P, glucose-6-phosphate; LDH, lactate dehydrogenase; MCT, monocarboxylic acid transporter; NADPH, nicotinamide adenine dinucleotide phosphate; PPP, pentose phosphate pathway; R5P, ribulose-5-phosphate; TCA, tricarboxylic acid.

**Figure 2 ijms-23-03785-f002:**
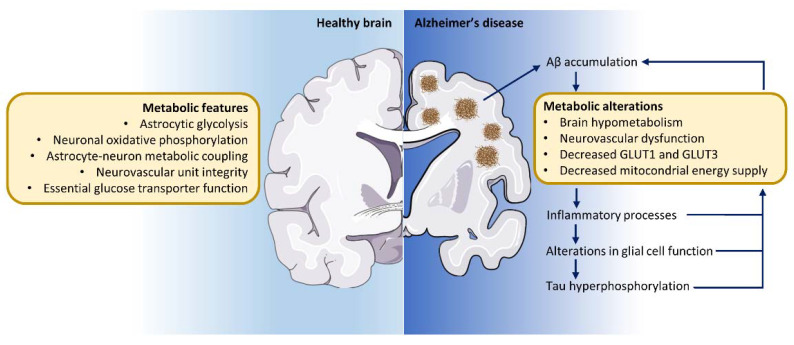
Metabolic alterations contribute to Alzheimer’s disease pathology. The metabolic coupling between astrocytes and neurons sustains proper brain function in the healthy brain. In the course of Alzheimer’s disease, at least 15 years before the onset of symptoms, a marked glucose hypometabolism is detected in specific brain regions. The accumulation of Aβ exacerbates brain glucose hypometabolism, both directly within the area of Aβ accumulation as well as in remote regions. In turn, this hypometabolic state triggers inflammatory processes and cellular damage. Moreover, alterations in glial cell function together with tau phosphorylation and subsequent accumulation exacerbates brain metabolic breakdown. Together, these features perpetuate a vicious cycle of neurodegeneration and declining brain glucose metabolism that contributes not only to the deterioration of memory and cognition but also to abnormal behavior in affected patients.

**Figure 3 ijms-23-03785-f003:**
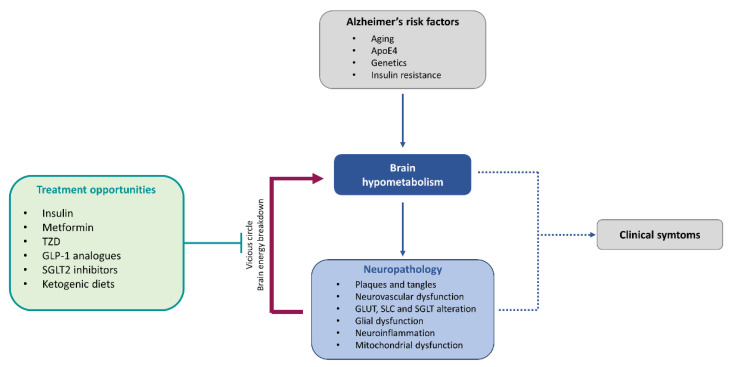
Central contribution of brain energy hypometabolism in Alzheimer’s disease pathology and treatment opportunities. Glucose hypometabolism occurring at early stages of the disease, together with the neuropathological features, induces a vicious cycle leading to brain energy breakdown and dysfunction. Treatment opportunities based on energy rescue strategies attempt to break this vicious circle and improve brain energy metabolism to, in last instance, prevent or revert clinical symptoms.

**Table 1 ijms-23-03785-t001:** The facilitative glucose transporters (GLUTs) and sodium-dependent glucose co-transporters (SGLTs) in the human nervous system.

	Gene	Transporter	Anatomical Location	Cellular Location	Proposed Function	Glucose Affinity (mM)
Facilitative, sodium-independent	*SLC2A1*	GLUT1	Brain	Astrocytes, endothelial cells	Main mediator of brain glucose uptake	1–2
	*SLC2A2*	GLUT2	Brainstem, hypothalamus	Astrocytes, neurons, tanicytes	Glucose sensor	15–20
	*SLC2A3*	GLUT3	Brain	Neurons, astrocytes	Direct neuronal glucose uptake	1–2
	*SLC2A4*	GLUT4	Cerebellum, cortex, hippocampus, hypothalamus, pituitary	Astrocytes, neurons	Fueling of active synapses	5
	*SLC2A5*	GLUT5	Brain	Microglia	Fructose transport, microglial-specific glucose transport	—
	*SLC2A6*	GLUT6	Brain	Neurons	—	5
	*SLC2A7*	GLUT7	Brain	Astrocytes	Glucose supply from astrocytes to other brain cells	—
	*SLC2A8*	GLUT8	Amygdala, cerebellum, hippocampus, hypothalamus	Neurons	Hippocampal neurogenesis, intracellular glucose transport	2
Sodium-glucose co-transport, sodium-dependent	*SLC5A1*	SGLT1	Brain	Endothelial cells, astrocytes, neurons	Brain glucose uptake under oxygen/glucose deprivation	0.2
	*SLC5A2*	SGLT2	Brain	Endothelial cells	Brain glucose uptake under oxygen/glucose deprivation	10
	*SLC5A4*	SGLT3	Brain	Neurons	Glucose sensor without glucose transport	2

**Table 2 ijms-23-03785-t002:** MCTs in the human nervous system.

Gene	Transporter	Anatomical Location	Cellular Location	Proposed Function	Lactate Affinity (mM)
*SLC16A1*	MCT1	Brain	Endothelial cells, astrocytes	Lactate efflux from glycolytic cells	3.5–10
*SLC16A7*	MCT2	Brain	Neurons	Neuronal lactate influx	0.5–0.75
*SLC16A8*	MCT3	Retina, choroid plexus	Oligodendrocytes, neurons	Subretinal space pH regulation	5–6
*SLC16A3*	MCT4	Brain	Astrocytes	Lactate efflux from glycolytic cells	22–28

**Table 3 ijms-23-03785-t003:** Clinical and pre-clinical metabolic-related intervention for the treatment of AD.

Treatment	Type of Study	Drug	Observed Effects	References
**Insulin**	Pre-clinical	IN insulin	Improved memory, decreased amyloid plaques, decreased microglial activation, increased hippocampal neurogenesis.	[275,276,277]
	Clinical	IN insulin	Improved delayed and working memory, improved verbal information, increased plasma amyloid levels, preserved brain glucose uptake and preserved brain volume.	[278,279,280,281,282]
		Long acting insulin	No beneficial effects.	[283,284]
		Fast acting insulin	No beneficial effects, stopped Phase II.	[285]
**Metformin**	Pre-clinical	Metformin	Increased mitochondrial function, neuroprotection, cognition.	[286,287,288,289,290]
	Clinical	Oral Metformin	Improved executive function, learning and verbal memory. No effect of Aβ or Tau.	[261,262]
		Long-acting Metformin	Recruting phase II/III.	
**TZD**	Pre-clinical	Rosiglitazone	Improved spatial memory, decreased Aβ and Tau brain burden.	[291]
		Pioglitazone	No effect on memory, Aβ or Tau. Decreased oxidative stress and astrocyte activation. Increased brain glucose uptake.	[292]
	Clinical	Rosiglitazone	Improved memory. No changes on Aβ.	[266]
			No beneficial effects.	[293]
		Pioglitazone	Improved cognitive scores. Increased cerebral blood flow. No changes on Aβ.	[265]
			No beneficial effects, stopped Phase III	[263]
**GLP-1 analogues**	Pre-clinical	Liraglutide	Improved learning and memory. Decreased microglial activation. Improved neurogenesis.	[294,295,296]
		Exendin-4	Prevents memory impairment. Decreased Tau.	[297]
	Clinical	SC Liraglutide	No effect on cognition. Prevents decline in brain glucose uptake.	[269,270,272]
		Exendin-4	No effect on memory, Aβ or Tau.	[271]
		Dulaglutide	Decreased cognitive impairment.	[268]
**SGLT2 inhibitors**	Pre-clinical	Empagliflozin	Decreased memory impairment, neuronal loss, oxidative stress and vascular dysfunction. Improved glucose metabolism.	[201,298]
		Dapagliflozin	Decreased memory impairment and oxidative stress. Improved brain insulin sensitivity and synaptic plasticity.	[200]
	Clinical	Empaglifozin	Recruiting Phase I	
		Dapagliflozin	Recruiting Phase I/II	
**Ketogenic diet**	Pre-clinical	Ketone ester	Increased glycolysis, mitochondrial functions, cognition, motor performance. Decreased Aβ and Tau levels.	[299,300,301,302,303]
	Clinical	Low carbohydrate ketogenic diet	Improved memory and cognitive functions.	[304,305,306,307,308,309,310,311,312]

Abbreviations: Aβ, β-amyloid; GLP-1, glucagon like-peptide 1; IN, intranasal; SGLT2, sodium-glucose co-transporter 2; SC, subcutaneous; TZD, thiazolidinediones.

## Data Availability

Not applicable.

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
