# Peer review of "Brain Metabolic Alterations in Alzheimer’s Disease"

_ijms, 2022, doi:10.3390/ijms23073785_

Round 1
Reviewer 1 Report
It is a well written manuscript and topic is very important, but there are some minor points that author could improve to increase quality.
- Author described too much about energy metabolism process in brain, which is too general and does not comply with the title. As the title claim metabolic alterations in AD, author should describe more connection between metabolic signaling in AD pathogenesis and include one more figure.
- Please include microglial scenario, how does metabolic impairments affect microglial environment is necessary for this article.
- Section 2 & 3 is important but are not the main point under this title, please concise those and discuss elaborately about metabolic disorders and how does that affect AD pathogenesis.
- Section 5 is the key to this article but it is written in superficially
- Please include future directions to study metabolic changes in progression or pathogenesis of AD.
- Please add therapeutic importance of metabolic homeostasis to prevent or halt AD.
Author Response
It is a well written manuscript and topic is very important, but there are some minor points that author could improve to increase quality.
We thank the reviewer its helpful comments to increase the quality of the paper.
- Author described too much about energy metabolism process in brain, which is too general and does not comply with the title. As the title claim metabolic alterations in AD, author should describe more connection between metabolic signaling in AD pathogenesis and include one more figure.
We thank the reviewer to point this issue to our attention. Section 5 (Energy metabolism alterations in Alzheimer’s disease) has been profoundly changed and extended (please see new paragraphs marked in blue) in order to offer more evidences of the connection between metabolic changes and AD pathogenesis.
A new figure connecting brain hypometabolism, AD neuropathological markers and treatment opportunities has been added to the manuscript (please see lines 669-689).
- Please include microglial scenario, how does metabolic impairments affect microglial environment is necessary for this article.
The link between metabolic alterations and microglia has been included in section 5 (please see lines 606-628).
- Section 2 & 3 is important but are not the main point under this title, please concise those and discuss elaborately about metabolic disorders and how does that affect AD pathogenesis.
With the extension of section 5 and the addition of a new section (Section 6), we hope the reviewer will find now those sections more balanced with sections 2 and 3.
- Section 5 is the key to this article but it is written in superficially
Section 5 has been profoundly changed and extended. We hope the information is now more complete.
- Please include future directions to study metabolic changes in progression or pathogenesis of AD.
Two new paragraphs englobing future directions have been included inside section 6.
- Please add therapeutic importance of metabolic homeostasis to prevent or halt AD.
A new section concerning treatment opportunities based on metabolic modifications has been added to the paper (please see lines 655-717). We hope this section can illustrate the metabolic modification options for a future treatment for AD.
Reviewer 2 Report
Line 115-134: need a table to summarize the distribution and roles of GLUTs
Line 145: SLC: fullname The solute carrier (SLC)
Line 146: SGLT fullname : (the Na+-coupled glucose transporters (SGLT))
Line 151-160: need a table to summarize the distribution and roles of MCTs,
Line 376:AD, PD, ALS, HD, fullname
- Energy metabolism alterations in Alzheimer’s disease
This section is too complicate, not easy to be understood.
Apart from GLUTs, are the SLC or SGLT involved in the AD?
Please see the reference https://link.springer.com/article/10.1007/s00424-020-02441-x
https://pharmrev.aspetjournals.org/content/72/1/343
Author Response
We thank the reviewer all the suggestions to improve the paper.
Line 115-134: need a table to summarize the distribution and roles of GLUTs
A table summarizing this information has been added. Please see lines 152-172.
Line 145: SLC: fullname The solute carrier (SLC)
The fullname has been included in lane 108 (please see the text highlighted in blue).
Line 146: SGLT fullname : (the Na+-coupled glucose transporters (SGLT))
The fullname has been included in lane 146 (please see the text highlighted in blue).
Line 151-160: need a table to summarize the distribution and roles of MCTs,
A table summarizing this information has been added. Please see lines 187-195.
Line 376:AD, PD, ALS, HD, fullname
The full term is previously cited in the text (please see, line 60-62).
- Energy metabolism alterations in Alzheimer’s disease
This section is too complicate, not easy to be understood.
The complete section has been changed and extended. We hope the reviewer will find this section clearer now.
Apart from GLUTs, are the SLC or SGLT involved in the AD?
We thank the reviewer to point this issue to our attention. Two paragraphs in section 5 have been included concerning the involvement of SLC and SGLT in AD.
Reviewer 3 Report
The authors are dealing with very important Brain alterations in Alzheimer’s disease and they have stated that several studies have evidenced that metabolic alterations could be in the center of the onset and progression of many neurodegenerative diseases. Thus, aging processes induces decreases in the glucose and oxygen metabolic rates of the brain and this metabolic breakdown could be further exacerbated in degenerative disorders such as the Alzheimer’s disease (AD),Parkinson’s disease(PD),Huntington’s disease(HD) and amyotrophic lateral sclerosis(ALS). The authors are stated that aging is an irreversible process and is characterized by the time-dependent functional decline of physiological integrity and also is the main risk factor for the neurodegenerative diseases for millions people aged 65 years or over. In addition the authors have stated plethora of the senescent brain alterations which are related to aging but are not completely understood at the molecular level. Another point that the authors include in this research is that the brain suffers from a marked decrease
in energy metabolism during senescence. In this respect, glucose hypometabolism and mitochondrial dysfunction could be measured by neuroimaging techniques. The authors also have stated remarkably that the anatomical distribution of deficient brain glucose metabolism serves as a clinical tool to distinguish AD from fronto-temporal dementia(FTD),PD,Leawy body disease and other types of dementia.
In addition ,the authors have stated that the age for AD patients is larger that 65 years but the large majority (90-95%) has a late onset.
Although ,the etiology of the AD disease is still unknown, the pathological hallmarks are well described and the AD is characterized by the presence of amyloid plaques , made up of extracellular amyloid-β(Αβ) peptide, and neurofibrillary tangles,
composed by intracellular hyperphosphorylated tau protein. The degree of the Αβ clearance has a great role in its accumulation in the brain.The authors also stated that the AD can be considered as a proteinopathy where peptides are deposited in the brain structures and therefore are critical for memory and cognition. The authors also concluded that this induces loss of neuronal functions and alteration of synaptic connections and neuronal cell death in different brain regions. Among the most affected areas is the hippocampus which it is widely accepted as the declarative memory region.
From all of the above mentioned the authors have concluded that there is no effective treatment or cure for AD. Several drugs can ameliorate symptoms and retard cognitive decline for 6-12 months but none of them stops the progression and also most of the AD new therapies have failed to revert the symptoms. The authors taking into account the variety of factors which altered the brain energy metabolism in AD, this approach they considered to be clinically ineffective and multimodal intervention could be more effective for successful brain energetic rescue. Finally, they have concluded that maintaining the energy status of the brain should become the cornerstone for trials attempting to delay the onset and progression of the AD.
Therefore, taking into my review all of the above citations of the authors I have accepted this paper for publication in your journal.
Author Response
Therefore, taking into my review all of the above citations of the authors I have accepted this paper for publication in your journal.
We thank the reviewer for the acceptance of the paper.